# Roles of Glutamate Receptor-Like Channels (GLRs) in Plant Growth and Response to Environmental Stimuli

**DOI:** 10.3390/plants11243450

**Published:** 2022-12-09

**Authors:** Bo Yu, Nian Liu, Siqi Tang, Tian Qin, Junli Huang

**Affiliations:** Key Laboratory of Biorheological Science and Technology of Ministry of Education, Bioengineering College, Chongqing University, Chongqing 400044, China

**Keywords:** glutamate receptor-like channels (GLRs), Ca^2+^, growth and development, environmental stress response

## Abstract

Plant glutamate receptor-like channels (GLRs) are the homologues of ionotropic glutamate receptors (iGluRs) that mediate neurotransmission in mammals, and they play important roles in various plant-specific physiological processes, such as pollen tube growth, sexual reproduction, root meristem proliferation, internode cell elongation, stomata aperture regulation, and innate immune and wound responses. Notably, these biological functions of GLRs have been mostly linked to the Ca^2+^-permeable channel activity as GLRs can directly channel the transmembrane flux of Ca^2+^, which acts as a key second messenger in plant cell responses to both endogenous and exogenous stimuli. Thus, it was hypothesized that GLRs are mainly involved in Ca^2+^ signaling processes in plant cells. Recently, great progress has been made in GLRs for their roles in long-distance signal transduction pathways mediated by electrical activity and Ca^2+^ signaling. Here, we review the recent progress on plant GLRs, and special attention is paid to recent insights into the roles of GLRs in response to environmental stimuli via Ca^2+^ signaling, electrical activity, ROS, as well as hormone signaling networks. Understanding the roles of GLRs in integrating internal and external signaling for plant developmental adaptations to a changing environment will definitely help to enhance abiotic stress tolerance.

## 1. Introduction

Plant glutamate-like channels (GLRs) are homologues of ionotropic glutamate receptors (iGluRs), which are nonselective ligand-gated cation channels in the nervous system of mammals and mediate the most excited synaptic signals between neurons [1]. The mammal iGluR family consists of complex allosteric proteins that change conformation by binding to the neurotransmitter glutamate, leading to the opening of transmembrane pores through which ions can flux [2]. Generally, the iGluR family includes three major subtypes: α-amino-3-hydroxy-5-methyl-4-isoxazole propionic (AMPA), kainite (KA), and N-methyl-D-aspartate (NMDA) receptors [1], which share common structural features but have diverse kinetic and pharmacological properties and different functions in synaptic transmission, learning, memory formation, and brain development [3]. Plant GLR homologues were first reported in Arabidopsis (*Arabidopsis thaliana*), and twenty GLR members were grouped into three clades [4,5]. Evolutionary insight into plant GLRs over the entire plant timescale showed that tandem duplications occupied the largest proportion of the GLR gene family expansion and also identified unique targets for manipulation of the woody-growth behaviors of GLRs [6,7]. As a highly conserved family of ligand-gated ion channels, plant GLRs have the same conserved primary domain as *Escherichia coli* glutamine permease (GlnH) and animal iGluRs, including the ‘three-plus-one’ transmembrane domains (M1 to M4) and the putative ligand-binding domains (GlnH1 and GlnH2) [4], and these findings lay a solid foundation for studying and elucidating the functions of GLRs in plants [8]. Plant GLRs are similar to iGluRs in channel properties but have a distinct symmetry, inter-domain interfaces, ligand specificity, and a non-swapped domain arrangement [8,9,10,11]. Notably, plant GLRs have been shown to be involved in various Ca^2+^-mediated developmental processes and physiological responses, including the response to light [4], the spontaneously and chemically-elicited electrical activity of the root apex [12], pollen tube growth [13], microtubule-mediated aluminum-sensitivity [14], and wound response [15].

Despite recent progress in the roles of the plant GLR family, limited knowledge concerning the biochemical properties of GLRs is understood, and the complex physical interaction and coordination between GLRs and other proteins located on the plasma membrane still need to be further explored. This review summarizes recent progress in the roles of GLRs in plant growth and development and responses to environmental stimuli, which will surely fuel future research on the many unanswered questions about GLRs that plant biologists have been interested in for decades.

## 2. GLR-Mediated Ca^2+^ Signaling Regulates Various Plant Physiological Processes

As extracellular amino acid sensors, plant GLRs play fundamental roles in regulating various physiological processes, which are closely associated with Ca^2+^ signaling [16,17,18,19]. With effective amino acids, plant GLRs channel many kinds of cation fluxes across the membrane into the cytoplasm, in particular Ca^2+^, which acts as a main signal messenger [20,21]. Arabidopsis *GLR3.1* (*AtGLR3.1*) is preferentially expressed in stomatal guard cells, and the overexpression of *AtGLR3.1* impairs stomatal closure, suggesting that AtGLR3.1 is closely correlated with the cytosolic Ca^2+^ concentration ([Ca^2+^]_cyt_) that regulates stomatal movement [22]. As a fact, convincing evidence has been presented to confirm the Ca^2+^ permeability of GLRs, which have a broad agonist profile. Studies of Arabidopsis hypocotyl cells indicated that six effective amino acids trigger transient Ca^2+^ influx and membrane depolarization by a mechanism that depends on AtGLR3.3 [23], and rice GLR3.4 (OsGLR3.4) has a broad agonist profile with eleven amino acids that induce transient Ca^2+^ influx in an OsGLR3.4-dependent manner in coleoptile epidermal cells [19]. AtGLR3.4-mediated Ca^2+^ signaling was involved in the regulation of seed germination under salt conditions through the SOS pathway, and the *atglr3.4* mutant showed impaired [Ca^2+^]_cyt_ induction as well as a reduced expression of *ABSCISIC ACID-INSENSITIVE* (*ABI*) genes, *AtABI3* and *AtABI4*, in response to salt stress [20,24]. Similarly, AtGLR3.5-mediated [Ca^2+^]_cyt_ enhancement promotes seed germination by counteracting the inhibitory effects of ABA through the repression of *ABI4* expression [25]. Consistently, applying exogenous glutathione to Arabidopsis leaves triggered a transient rise in [Ca^2+^]_cyt_, but this response was impaired in the *atglr3.3* mutant [13]. In another report, AtGLR1.2 channeled Ca^2+^ influx into the pollen tube cells when they extended in the pistil, but Ca^2+^ signaling was largely impaired in *atglr1.2* pollen tubes [26]. In brief, these plant-specific physiological processes are regulated by GLR-mediated Ca^2+^ signaling (Figure 1). Understanding of the roles of GLRs in plant growth and abiotic stress will help with the engineering of perfect-fitness and stress-tolerant crops.

## 3. Roles of GLRs in Plant Growth and Development

### 3.1. Roles in Seed Germination

Seed germination is a major step in plant growth, and it is strictly controlled by endogenous and environmental signals such as phytohormones and environmental factors including water, temperature, and light [27]. ABA is known to regulate the sophisticated process of seed maturation and germination [28]. Recent studies indicated that GLRs are involved in the seed germination by modulating the levels of ABA and ethylene in Arabidopsis [29,30]. Carbon (C) and nitrogen (N) are the two most critical elements that are required for normal plant development and metabolism [31]. Plants need to coordinate C and N metabolism to control their growth and development during different periods, and they therefore have evolved a sophisticated regulatory system to continuously monitor the levels of different carbon–nitrogen ratio ‘‘check point’’ molecules, including sucrose (Suc), glucose (Glc), 2-oxoglutarate, glutamine (Gln), glutamate (Glu), NO_3_^−^, and NH_4_^+^ [29,32]. AtGLR1.1 is involved in C and N metabolism and controlling seed germination by affecting ABA levels [29]. During seed germination, the addition of NO_3_^−^ can stimulate the formation of Glu and Gln, which activate the expression of *AtGLR1.1* and ultimately inhibit ABA synthesis; alternatively, the transcription of *AtGLR1.1* can be repressed by Suc, which induces the expression of ABA biosynthetic genes and thus leads to ABA-mediated seed germination inhibition [29]. Despite the implications that AtGLR1.1 might function as a C/N receptor or sensor, the true ligand of AtGLR3.1 is yet unclear, and this needs to be experimentally investigated. In other reports, transcription factors associated with the ABA signaling pathway, AtABI3, AtABI4, and AtABI5, were found to have important regulatory effects on seed maturation and germination, among which ABI3 was essential for seed maturation and desiccation tolerance, while ABI4 and ABI5 played a crucial role in ABA inhibition during seed germination [33,34,35,36]. It has been shown that there is a large amount of Ca^2+^ in the seed coat, cell wall, and apoplast [37,38]. Extracellular Ca^2+^ influx can promote seed germination, but the process can be remarkably inhibited when this influx is disturbed, and the exogenous application of Ca^2+^ can attenuate ABA inhibition during seed germination [25,39]. *AtGLR3.5* is mainly expressed in germinated seeds, and the AtGLR3.5-mediated [Ca^2+^]_cyt_ elevation can alleviate the inhibition of ABA during seed germination, demonstrating the essential roles of AtGLR3.5 in seed germination [25]. Compared to the control, the transcript abundance of *AtABI4* was significantly increased in germinating *AtGLR3.5*-RNAi seeds, whereas it was reduced in *AtGLR3.5*-overexpression seeds, and the changes in germination were consistent with the alteration of [Ca^2+^]_cyt_ [25]. This observation demonstrates that the AtGLR3.5-mediated increase in [Ca^2+^]_cyt_ promotes seed germination due in large part to the decrease in the *ABI4* expression. The application of Glu can reduce the inhibiting effect of salt stress on Arabidopsis seed germination [40], suggesting that the GLR-mediated [Ca^2+^]_cyt_ elevation may alleviate the inhibition caused by environmental stress during seed germination. Presently, seeds face challenges in viability. Seed dormancy is a key characteristic which inhibits seed germination during the harsh and tough growing season. Due to the crucial roles of ABA in dormancy induction and maintenance in the adverse environment, GLRs have the potential to improve seed vigor by restricting ABA signaling and thus are expected to be used for genetic modifications in crops. Therefore, the identification of GLRs as potential players in seed germination facilitates their practical use in breeding to attain a high capacity of seed germination and seedling establishment, especially under salinity stress.

### 3.2. Roles in Root Development

Plant root system (including primary roots and lateral roots) development is a highly ordered process [41,42], which is critical for efficient nutrient uptake, drought resilience, and crop yield [43]. During root development, the maintenance of cell division and individual cell survival in the root apical meristem are correlated with spatiotemporal characteristics of the electrical network activity of the root apex [12,42]. The Arabidopsis *atglr3.6-1* mutant has a reduced mitotic activity and root meristem size in the root tip, which results in shorter primary roots and fewer lateral roots, while *AtGLR3.6* overexpression stimulates both primary and lateral root development [44]. The cyclin-dependent kinase (CDK) inhibitor Kip-related protein 4 (KRP4) is an inhibitor of the cell cycle [45,46]. Accordingly, the transcript abundance of *AtKRP4* in *atglr3.6-1* roots was significantly enhanced, while reduced in transgenic lines overexpressing *AtGLR3.6* [44], suggesting that AtGLR3.6 plays a positive role in maintaining root meristem by repressing the expression of *AtKRP4*. Consistently, [Ca^2+^]_cyt_ was greatly reduced in the root meristem of *atglr3.6-1*, and the addition of calcium reduced the expression of *AtKRP4* and promoted root growth [44]. Members of the PIN (PIN-FORMED) family are responsible for polar auxin transport in plants [47]. In *atglr3.6-1* roots, *AtPIN1* mRNA was less abundant, and auxin levels were lower, indicating that the downregulation of the *AtPIN1* expression could be the cause of the reduced auxin level [44]. The reduction in auxin levels in *atglr3.6-1* roots resulted in a reduced mitotic activity in the root tips [44], which implies that AtGLR3.6 is involved in PIN-mediated auxin polar transport by mediating Ca^2+^ influx. In addition, AtGLR3.2 interacts with AtGLR3.4 to form the heteromeric AtGLR3.2/AtGLR3.4 channel, and the single mutants *atglr3.2* and *atglr3.4* as well as the double mutant *atglr3.2**atglr3.4* display an equally severe phenotype, a large overproduction, and aberrant placement of lateral root primordial [48], suggesting they play essential roles in lateral root development via Ca^2+^ signaling. In addition to cell division and differentiation, the fundamental development process of root development is also accompanied by programmed cell death, including aerenchyma formation, root cap cell production, and shedding [49,50], suggesting that root development is a well-coordinated program regulated by multiple factors. Research on the nervous system in mammals has shown that iGluRs are vital in the fate determination of nerve cells at the early stage of development [51]. In the rice *Osglr3.1* mutant, cell division and differentiation in the root apex were seriously affected, and the programmed cell death of root meristem was increased, resulting in the inhibition of primary root elongation [42], suggesting that plant GLRs might have similar roles to those in their animal homologues. It is recognized that treating Arabidopsis roots with Glu elicits rapid changes in the membrane potential and increases Ca^2+^ flux, and specific members of the GLR family are required for this response [23,52]. Similarly, our recent study showed that the application of Glycin (Gly, one agonist of OsGLR3.4) inhibited primary root growth in rice, while *Osglr3.4* mutants exhibited hyposensitivity to the exogenous Gly treatment. The mechanistic study showed that Gly treatment triggered OsGLR3.4-mediated [Ca^2+^]_cyt_ elevation, which subsequently induced membrane depolarization and ROS production in the root tips [53]. Despite the genetic evidence about the roles of GLRs in root growth, we still largely misunderstand how GLRs act in the coordination of primary root growth and lateral root proliferation. Improving crop root architecture and function may be challenging, but a series of cases have emerged showing that genetic modifications of roots, when the modified gene is specifically expressed in the root, result in enhanced plant performance, increased yield, and elevated stress tolerance. Therefore, the prominent roles of GLRs in root development make them potential candidates for increasing crop production. Additionally, more work is needed to explore the role of different GLR isoforms in root growth.

### 3.3. Roles in Other Developmental Programs

Pollen tubes are a model system for studying tip growth, which involves many parameters including vesicle trafficking, cell wall precursor exocytosis, actin microfilament polymerization, apical ion flux, and [Ca^2+^]_cyt_ oscillation [54]. Arabidopsis *atglr1.2* and *atglr3.7* knockout plants displayed male reproductive phenotypes, and further mechanistic investigation demonstrated that AtGLR1.2 and AtGLR3.7 could facilitate Ca^2+^ influx across the plasma membrane, modulate the apical [Ca^2+^]_cyt_ gradient, and consequently affect pollen tube growth and morphogenesis [26]. As a fact, some GLRs are able to function in a series of programs related to plant growth and development. For instance, in addition to the modulation of primary root growth, new roles for OsGLR3.4 were found in the development of rice plant architecture. *Osglr3.4* mutants showed a semi-dwarf phenotype with erect leaves, and OsGLR3.4 was shown to be involved in brassinosteroid (BRs) signaling to modulate cell elongation of the internode and lamina joint [19]. As a Ca^2+^-permeable channel activated by multiple amino acids, OsGLR3.4 modulates actin filament organization and vesicle trafficking by mediating the [Ca^2+^]_cyt_ rise, which is required for cell elongation [19]. More recently, we found that OsGLR3.4 modulated root tropism growth towards amino acids via plasma membrane depolarization and ROS generation [53]. Excitingly, OsGLR3.4 was demonstrated to promote nitrate uptake by modulating the Gly-induced expression of nitrate transporter genes [53]. Therefore, OsGLR3.4 is a potential candidate for plant adaption to an uneven nutrient distribution as well as to promote nitrate uptake, which will facilitate the practical use of OsGLR3.4 in rice molecular breeding for a high crop yield.

## 4. Roles of GLRs in Plant Response to Environmental Stress

Plants are subjected to a series of environmental stresses during their growth and development, and they often display considerable plasticity in their developmental and physiological behaviors by responding to a variety of environmental signals, including mechanical wounding, unexpected herbivory attack, water deficit, and soil salinity [15,55,56]. It has been recognized that GLRs play a vital role in environmental signal reception and transmission and plant adaption to environmental stress, including biotic and abiotic stresses [20,57,58,59,60]. Recent progress regarding the functions of GLRs in plant response to environmental stress is discussed in detail below.

### 4.1. Roles in Mechanical Wounding or Herbivory Attack

Accumulating evidence has demonstrated that GLR-mediated Ca^2+^ signaling is rapidly activated in response to touch or mechanical wounding in plants [61] (Figure 2). Studies have shown that when Arabidopsis is exposed to touch or cold, the transcripts of *AtGLR3.4* increase significantly, suggesting that *AtGLR3.4* is involved in the response to these environmental stimuli [61]. In addition, GLRs were shown to play essential roles when plants were subjected to herbivory attack. Unlike animals [62], plants do not have specialized nerve cells with axons, and they cannot rely on a rapid nervous system to avoid dangerous threats in the environment [15]. However, similar to animals, they do operate long-distance electrical signals [63]. By using noninvasive electrodes to detect changes in the membrane potential in Arabidopsis leaves, researchers found that membrane potential depolarization is correlated with the distal production of jasmonic acid (JA) in undamaged leaves [15]. When attacked by herbivores, plants suffer from mechanical wounding or invasion of exogenous chemicals, and they therefore generate electrical signals by activating GLRs [64,65]. These signals are then transferred to adjacent tissues where the biosynthesis of JA is induced, which in turn triggers the JA resistance pathway [66,67,68]. As expected, when mutations in *GLR* genes of clade 3 (*AtGLR3.2/3.3/3.6*) happen, the electrical signals transmitted to the adjacent leaves are attenuated in these mutants [15]. Further study indicated that wounding initiates the AtGLR3.3- or AtGLR3.6-dependent propagation of membrane depolarization in the form of short wave potentials (SWPs) that leads to defense gene activation, and *atglr3.3* and *atglr3.6* mutants were found to be compromised in their defense against herbivores [69,70]. It is notable that the signal transmission between leaves of the double mutant *atglr3.3atgrl3.6* decreased significantly, and the expression of JA-responsive genes was also reduced significantly in the leaves adjacent to the injured leaves [15]. The long-distance wounding response is also conserved in monocotyledon plant rice. Root injury triggering SWPs as well as the JA response in leaves are impaired in *Osglr3.4* mutants, indicating that OsGLR3.4 is required for root-to-shoot systemic wound signaling in rice [19]. Tomato SlGLR3.3- and SlGLR3.5-mediated leaflet-to-leaflet electrical signal transduction and herbivory-induced JA accumulation and *Helicoverpa armigera* resistance were reduced in *slglr3.3* and *slglr3.5* mutants [71], revealing the key roles of SlGLR3.3 and SlGLR3.5 in electrical signal transduction and JA signal activation. These studies provide a genetic basis for further study on the transmission mechanism of plant electrical signals between organs and also reveal some similarities in the transmission modes of electrical signals between plants and animals. Despite the genetic evidence about the role of GLRs in the local and long-distance transmission of electrical signaling, knowledge about how GLRs are activated in vivo in response to wounding is still limited, and more work is needed to elucidate the sophisticated mechanism.

Research on plant resistance to herbivory attacks found that GLRs have a critical role in plant immunity against aphids in leaves [72,73] (Figure 2). By using a fluorescent Ca^2+^ biosensor (GCaMP3) to monitor the real-time [Ca^2+^]_cyt_ dynamics in Arabidopsis leaves during a green peach aphid feeding, a strong fluctuation in [Ca^2+^]_cyt_ was detected when aphids’ probes penetrated into the epidermal and mesophyll cells in the leaves [73]. Further study showed that an unknown receptor cooperates with BRASSINOSTEROID INSENSITIVE-ASSOCIATED KINASE1 (AtBAK1) [74,75], which then activates the expression of *AtGLR3.3* and *AtGLR3.6* to transform signals during aphid feeding [73]. More recent research indicated that AtGLR3.3 and AtGLR3.6 are involved in the regulation of various metabolites locally and systemically, including amino acids, carbohydrates, and organic acids [76], which provides new insight into the function of AtGLR3.3 and AtGLR3.6 in mediating metabolites in local and systemic leaves under insect attacks and highlights their roles in regulating insect resistance in systemic leaves. As a fact, the triad of GLR-mediated Ca^2+^, ROS, and electrical activity has been implicated in the response to wounding in plants [77,78]. For example, upon wounding, systemic changes in the membrane potential, Ca^2+^, and ROS are coordinated by AtGLR3.3 and AtGLR3.6 [79]. Although there is a knowledge gap about plant GLRs that needs to be filled, a deeply conserved function for GLRs that links wounding perception to distal protective responses makes them promising candidates for genetic modification in engineering pest-resistance in crops. Nowadays, crop yield is under enormous pressure, which is caused by biotic stresses including pests. Therefore, genetic modifications by which the *GLR* genes are expected to be expressed at the appropriate time only when plants are attacked by pests would be effective for pest control but do not interfere with or negatively affect their physiological processes under normal growth conditions.

### 4.2. Roles in Drought Response

Exposure of plants to a water-limiting environment during the developmental stages appears to activate cascades of physiological and developmental changes [80]. Plant leaves can control the rate of water evaporation by adjusting the opening and closing of stomata, which is precisely regulated by ABA signaling [81,82,83]. When plants are exposed to drought stress, the increased ABA levels in the guard cells induce the accumulation of reactive oxygen species (ROS) produced by RbohD and RbohF NADPH oxidases [84,85]. The enhanced ROS activate the Ca^2+^ channels on the cell membrane, causing [Ca^2+^]_cyt_ elevation [86,87]. The increase in [Ca^2+^]_cyt_ eventually causes stomatal closure by activating *S*-type anion channels or inhibiting inward-rectifying K^+^ channels and H^+^-ATPases [88,89]. Recently, a new mechanism that differs from ABA-induced ROS production to stimulate stomatal closure was found [17]. L-methionine (L-Met) at the physiological concentration can increase [Ca^2+^]_cyt_ by activating AtGLR3.1 and AtGLR3.5, which eventually leads to stomatal closure [17]. The basal levels of [Ca^2+^]_cyt_ were significantly reduced in the single mutants *atglr3.1* and *atglr3.5* as well as the double mutant *atglr3.1atglr3.5*, and consequently, ABA-induced stomatal closure was also remarkably repressed in these mutants [17], suggesting that AtGLR3.1 and AtGLR3.5 might be involved in ROS-mediated stomatal closure and drought tolerance. It is notable that AtGLR3.5 was found to be the first cation channel located on the mitochondrial membrane. *AtGLR3.5* has two different splicing variants, and one of the variants targets the inner-mitochondrial membrane, while the other variant localizes to chloroplasts [90], suggesting an intricate mechanism that AtGLR3.5 is involved in to modulate stomatal movement. The ability of mitochondrial Ca^2+^ absorption was found to slightly decrease and the leaf showed obvious accelerated senescence in the *atglr3.5* defect mutant [90]. A study of *Medicago truncatula* showed that MtGLRs are required for adaptive responses under short-term water deficit stress during *Medicago* seedling establishment by mediating NO production [91]. Due to climatic variability, the incidence of drought stress at various crop growth stages is becoming a major hindering factor to yield improvement. Undoubtedly, knowledge about GLRs in plant drought response has important theoretical and practical significance for cultivating new drought-resistant crop varieties to enhance yield. Therefore, tight regulation and fine-tuning of *GLR* genes during plant stress responses will contribute to the establishment of complex signaling networks, and the important roles of GLRs in plant abiotic stress responses make them potential candidates for conferring stress tolerance.

### 4.3. Roles in Salinity Response

Soil salinization is an increasingly serious problem in global agriculture because excessive Na^+^ and Cl^−^ uptake by plant roots can disrupt metabolic processes and reduce photosynthetic efficiency and thus severely affect plant growth and development [92,93]. Generally, plants respond to osmotic stress by reducing water evaporation and maximizing water absorption [94]. In addition, plants can reduce the harmful effects of ion Na^+^ stress by excluding Na^+^ from leaf tissues and dividing Na^+^ into vacuoles [93,95]. Studies have shown that the sensitivity of the Arabidopsis *atglr3.4* mutant to NaCl was much higher than that of the wild-type plants [58], which implies that GLRs are required in the adaption to salinity stress. NaCl can significantly raise [Ca^2+^]_cyt_ in wild-type plants, and this response can be prevented by GLR antagonists; however, the increase in [Ca^2+^]_cyt_ induced by NaCl was significantly repressed in the *atglr3.4* mutant, and the expression of *AtSOS1*, *AtSOS2*, and *AtSOS3* in the mutant was also reduced [58]. Compared with that in the wild type, the Na^+^ content in the *atglr3.4* mutant was increased significantly [58], suggesting that AtGLR3.4-mediated Ca^2+^ signaling may regulate Na^+^ absorption through ROS signaling and may participate in the response to NaCl. Recently, AtGLR3.7 was reported to interact with 14-3-3 omega to modulate the salt stress response [96]. The mutant *atglr3.7-2* was more sensitive to salt stress, while *AtGLR3.7* overexpression lines exhibited the opposite trend [96].

When plants are subjected to salt stress, they are physiologically exposed to osmotic fluctuations, the toxicity of Na^+^ and K^+^, excessive ROS production, and unbalanced cytosolic K^+^ homeostasis [97]. Maintaining high K^+^ levels in plant cells has a positive effect on their response to salt stress, which allows vacuolar H^+^-PPase to maintain high activity and Na^+^ to be sequestered [98]. Plant root tips are more sensitive to salt stress than mature tissues, which is mainly attributed to the higher H^+^-ATPase activity in cells in mature tissues than in meristem [97], which might be associated with the fact that multiple *GLRs* are expressed in Arabidopsis roots [99,100]. As an activator of GLRs, the Glu gradient in the root tips is increased by about five times under salt stress [97]. It is estimated that Glu produced in the root tips under salt stress activates GLRs, which further activate the plasma membrane NADPH oxidase to produce excessive hydrogen peroxide (H_2_O_2_) [101], leading to the activation of outward-rectified K^+^ channels that cause programmed cell death [97]. On the contrary, the mature zones in roots do not produce excessive Glu under salt stress, and thus, the activation of GLRs is inhibited [97]. Ca^2+^ also acts as nutrition in plant growth and development, and a lack of Ca^2+^ leads to a series of physiological reactions in plants, including browning and death of the shoot apex, necrosis of leaf tips, and deformation of leaves [60]. The reduced growth resulting from calcium nutrition can also lead to enhanced sensitivity to salt stress in plants. The overexpression of *AtGluR2* (a homolog of the mammal ionotropic glutamate receptor gene) does not affect Ca^2+^ uptake but reduces its utilization, thus resulting in the phenotype of Ca^2+^ deficiency as well as hypersensitivity to Na^+^ and K^+^ stress [60]. *AtGluR2* is mainly expressed in vascular tissues, particularly in cells adjacent to conducting vessels, where Ca^2+^ in xylem sap is absorbed and distributed, and it is supposed to encode a functional channel that unloads Ca^2+^ from the xylem vessels into the cell membrane [60]. Thus, the appropriate expression of *AtGluR2* is required in Ca^2+^ nutrition by controlling the ion allocation among different Ca^2+^ sinks, either in normal development or adaptation to ionic stresses.

Plant defense against biotic and abiotic stresses generally comes at the expense of growth, and thereby, the tradeoff between defense and growth is a major constraint on plant evolution. Amino acids and their derivatives play vital roles in mediating defense priming and growth tradeoff [102]. Gated by amino acids and related molecules to induce Ca^2+^ signaling, GLRs are proposed to be potential molecules that induce defense priming and the equilibrium between growth and defense against stress including salinity, which is expected to help maintain relative plant fitness under unpredictable conditions and maximize reproductive success. Therefore, GLRs are promising candidates for genetic modification to improve plant tolerance to various stresses including salt stress and also provide an outlook on the prospects of engineering the tradeoff between defense and growth in plants.

### 4.4. Other Environmental Stimuli

In addition to herbivory attacks and osmotic stress, chilling is another major environmental stress that not only affects plant growth, but also minimizes the productivity and quality of crops [102,103]. SlGLR3.3 and SlGLR3.5 mediate the cold acclimation-induced chilling tolerance by regulating apoplastic H_2_O_2_ production and redox homeostasis [104]. Cold induces the expression of *SlGLR3.3* and *SlGLR3.5* in tomatoes coupled with an increased tolerance against subsequent chilling, while the silencing of *SlGLR3.3* or/and *SlGLR3.5* or the application of the antagonist of the ionotropic glutamate receptor compromises the cold-induced increase in the transcripts of *RBOH1* [104]. Aluminum is abundant in nature but harmful to plants, which largely limits the productivity of crops [14]. It was shown that the addition of aluminum could cause abnormal shapes of organs by depolymerizing root cell microtubules as well as depolarizing the plasma membrane, but the Ca^2+^ channel blockade can prevent these changes [14,105,106]. Plants have evolved sophisticated mechanisms to minimize or avoid aluminum toxicity, such as isolating aluminum in a stable form within cells or chelating them and making them harmless by excreting organic anions [107,108]. Root secretions are rich in glutamate and other amino acids, and the distribution of the amino acids secreted varies with the environment and development of the plants [109]. When plants are exposed to aluminum stress, the roots are stimulated to secrete organic acids, which are believed to be used for the activation of the plasma membrane anion channels including GLRs [14]. In support of this, a recent study on Arabidopsis showed that the Ca^2+^-dependent calmodulin-like protein CML24 interacts with CALMODULIN BINDING TRANSPORTER ACTIVATOR 2 (CAMTA2) and WRKY46 to regulate the ALUMINIUM-ACTIVATED MALATE TRANSPORTER 1 (ALMT1)-mediated secretion of malate from roots and consequently achieves aluminum tolerance [110].

Nitric oxide (NO) has been proven to be a key signal molecule in various physiological processes [111], including programmed cell death, growth, and the development of organs, seed germination, flowering, and response to biotic or abiotic stresses [112,113,114,115]. NO can be produced as the response of plants to different pathogen attacks and can promote the expression of defense-related genes, induce the formation of defense-related hormones, and ultimately is involved in the hypersensitivity reaction mechanism [116]. Studies have shown that, on one hand, NO affects the expression of genes encoding the CaM or Ca^2+^ channels, and on the other hand, directly or indirectly activates Ca^2+^ channels, the Ca^2+^/CaM-dependent protein kinase, and/or other Ca^2+^ sensors [59,117,118]. Other studies showed that, when plants are attacked by pathogens, the released Glu activates the GLR channel and causes cell membrane depolarization, allowing the signal to be transmitted to surrounding tissues [18,57]. The increased levels of GLRs promote the production of cryptogein (elicitor of the defense response) and NO [59]. Collectively, GLRs play an important role in the plant defense response by inducing NO production, although the mechanism still needs to be further explored. Based on the performance of GLRs during environmental stimuli, they are therefore considered to be promising candidates for crop breeding through genetic manipulations to increase the tolerance to biotic or abiotic stresses.

## 5. Interaction of GLRs with Hormone Signaling

For plants, both the growth and environmental response require rapid activation of Ca^2+^ signaling as well as hormone signaling (Figure 3). The tradeoff between growth and defense incorporates the balance between the plant ‘‘defense’’-related hormones, ABA, JA, and salicylic acid (SA), and those involved in growth, including BR, auxin, and gibberellic acid (GA) [119]. Accumulating evidence has revealed the central role of plant GLRs in the tradeoff between growth and defense. Our recent investigation showed *OsGLR3.4*, functioning as a target gene of transcription factor OsBZR1, is involved in BR-mediated plant height and architecture in rice [19]. Although the *AtGLR3.6* mutation leads to impaired [Ca^2+^]_cyt_ elevation coupled with a reduced root auxin level [44], which is tempting to speculate that the AtGLR3.6 interacts with auxin signaling, further work is needed to address the role of GLRs in auxin signaling. Research has shown that Ca^2+^ signaling integrates with GA signaling to contribute to plant growth and development, but there is little direct evidence that GLRs interact with GA signaling [120,121]. As for defense-related hormones, the interaction of GLR signaling with ABA and JA has been well discussed above, and here, GLRs interplaying with SA is focused upon. A recent study showed that plant GLRs work through SA signaling in their effects on tissue regeneration, and mutants of the SA receptor *NPR1* are hyper-regenerative and partially resistant to GLR perturbation, indicating that SA acts downstream of GLR signaling [122]. Meanwhile, the transcription of SA-responsive marker genes increases as cells mature, suggesting that SA mediates GLR signaling in older tissues and thus inhibits regeneration. Therefore, present evidence reveals the central role of GLRs in the tradeoff between wounding-triggered regeneration and defense, which offers new strategies to improve plant regeneration. It is important to note that several questions remain as to the molecular mechanisms underlying the modulation of GLRs on the tradeoff, and more work is required to elucidate how GLRs interact with hormone signaling to regulate the balance between plant growth and environmental response. Future studies dissecting the exact mechanisms underlying the dichotomous function of GLRs will undoubtedly provide even more tractable ways to increase plant growth as well as adaption to adverse environments.

## 6. Conclusions and Future Perspectives

Plant GLRs are a highly conserved membrane protein family, and they have overlapping expression patterns and biological functions. Present knowledge about the roles of GLRs in plant growth and response to environmental stress is mainly from the research on dicotyledon model Arabidopsis, whereas functions of most of the GLR members in monocotyledon plants have yet to be identified. Therefore, substantial experimental work is required to determine the specific biological function of GLRs from cereal crops, which will remain a substantial challenge in the coming years. Meanwhile, considering the large number of networks regulated by GLR-mediated Ca^2+^ signaling, it is believed that multiple unknown functions of GLRs remain to be explored. Recently, great progress on plant GLRs has been achieved through molecular genetics and electrophysiological methods. Despite the genetic evidence about the role of GLRs in the local and long-distance transmission of electrical signaling, the knowledge about the roles of different GLR isoforms in electrical signal transmission is still limited, and some open questions remain unanswered. For example, it is still unclear how the system’s electrical activity is propagated from the damaged tissues to the distal undamaged organs. Although long-distance electrical signals have been implicated to travel through vascular tissues, direct evidence is needed to answer this question, and future research will be directed toward addressing the question of plant long-distance signaling.

To achieve a better understanding of their roles during plant growth and environmental responses, it is essential to identify the interacting partners of GLRs that cooperate in regulating these physiological processes. It is also crucial to elucidate the effects of the interaction of GLRs with partners on their activation and reveal the possible molecular mechanism. Recent advances in elucidating the functions of Arabidopsis GLRs have revealed a mechanism for sorting and activation of Arabidopsis GLRs by CORNICHON HOMOLOG (CNIH) proteins [13]. CNIH proteins play an essential role in sorting, trafficking, and localizing GLRs, and more importantly, CNIHs can activate GLRs via the physical interaction between them [13]. Additionally, to gain a thorough understanding of how GLRs are activated in vivo, further studies on the mechanism are required, such as the possible role of CNIH in nonpollen tissues, the precise role of amino acid binding via the ligand-binding domain (LBD), and the significance of C-terminus phosphorylation. It is important to note that the interactions between GLRs and different partners may play different roles under different environmental stresses, such as drought, salt, cold stress, insect attacks, and pathogen infections. At present, only a few known interaction proteins of GLRs are characterized. The identification of interaction partners of GLRs would be helpful for understanding the detailed networks where GLRs function. Certainly, further molecular studies of GLRs will clarify the fine-tuned mechanisms that control Ca^2+^ signaling in plants, and it is likely to be crucial in helping to guide future breeding plans and consequently be beneficial for crop production.

## Figures and Tables

**Figure 1 plants-11-03450-f001:**
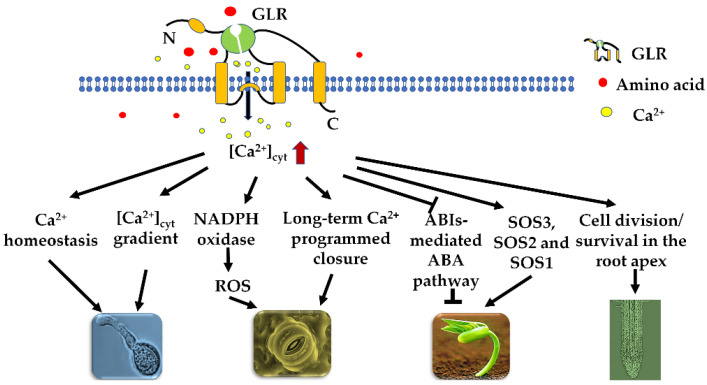
Summary of the main roles played by glutamate receptor-like channels (GLRs) in various physiological processes in plants. ABI, ABSCISIC ACID INSENSITIVE.

**Figure 2 plants-11-03450-f002:**
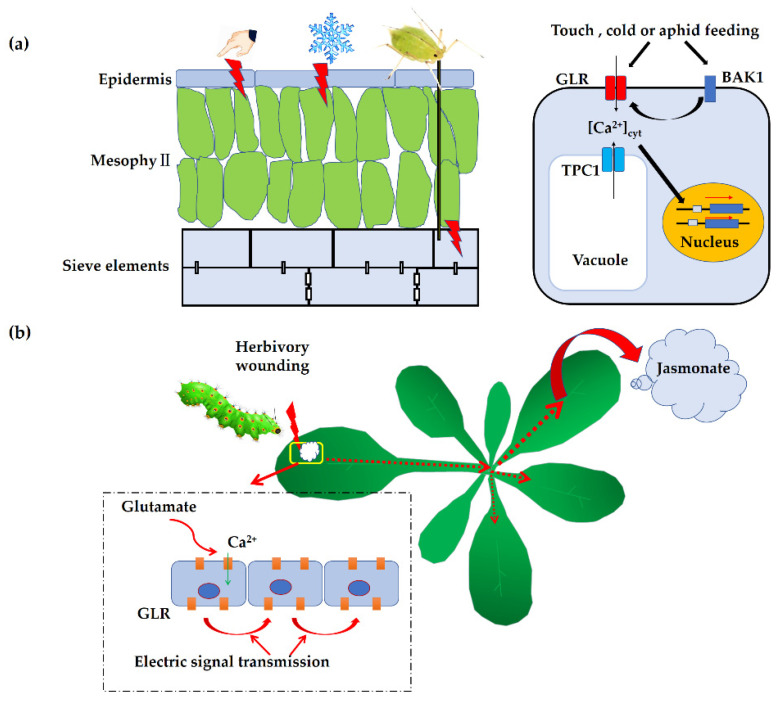
Plant glutamate receptor-like channels (GLRs) are involved in the response to environmental stimuli. (**a**) Touch, cold, or aphid feeding induces GLR-mediated signal transduction. (**b**) The role of GLRs in plant defense against herbivory wounding. TCP1, two-pore channel 1. BAK1, BRASSINOSTEROID INSENSITIVE-ASSOCIATED KINASE1.

**Figure 3 plants-11-03450-f003:**
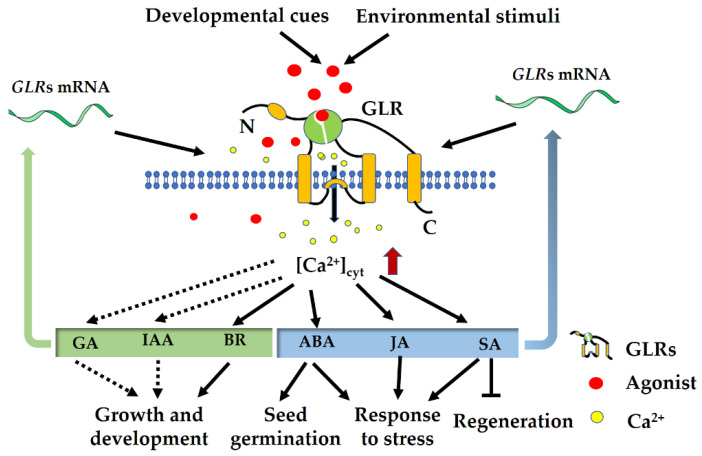
Glutamate receptor-like channel (GLR)-mediated Ca^2+^ signaling regulates hormone signaling to regulate plant growth and stress response. GA, gibberellic acid. IAA, indole-3-acetic acid. BR, brassinosteroid. ABA, abscisic acid. JA, jasmonic acid. SA, salicylic acid.

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
