# Peer review of "Roles of Glutamate Receptor-Like Channels (GLRs) in Plant Growth and Response to Environmental Stimuli"

_plants, 2022, doi:10.3390/plants11243450_

Round 1
Reviewer 1 Report
Title of the manuscript: Roles of Glutamate Receptor-like Channels (GLRs) in Plant Growth and Response to Environmental Stimuli
General comments:
In the article entitled "Roles of Glutamate Receptor-like Channels (GLRs) in Plant 2 Growth and Response to Environmental Stimuli", the authors reviewed the roles of GLRs in plants. They have listed the various roles of GLRs in plants with reference citations. But this manuscript reads as a compilation or catalog of facts without their critical analysis, thus providing no new ideas about the exploitation of GLRs to improve the agronomic traits of plants. There is a lot of opportunity to draw inroads with genetic tools like CRISPR/Cas to manipulate the GLR expression to improve the plant performance under adverse conditions, which authors didn't utilize in this manuscript.
Comments:
1. An image illustrating the GLR-mediated Ca2+ signaling might benefit the readers.
2. Lane no 91: In the sentence "...involved "in" the seed.... "in" is missing!
3. Lane no 111: Why are both "was" and "is"?? used in the sentence? The authors should correct it.
4. Lane no 113: Here and throughout the manuscript the authors should prefix the GLR names with specific plant names. For eg. AtGLA3.5, would help pinpoint the plant name without the need to write the entire plant name for each sentence. A space saving strategy!
5. Lane no 115: As asked above AtGLR3.1-RNAi?? Kindly specify the plant name for each GLR throughout the MS.
6. Lane no 87-120: Again this para merely compiles the past works without any critical analysis. Kindly add some critical insights on how germination could be increased by manipulating the expression levels of AtGLR3.5, etc.
7. Lane no 128: Is it AtGLR3.6??
8. Lane no136: "extracellular cells" doesn't seem to be a correct expression!
9. Lane no 146: Why so many "is"? in the sentence. Check and correct it.
10. Lane no 155-162: This para talks a lot about an electrical activity without a link to GLRs! Needs more clear rationale of GLRs here. Again the critical outlook is missing in root development section too.
11. Lane no 171: kindly change "osglr3.4" into "Osglr3.4". As asked above, like Osglr, prefix with the plant names for all GLRs.
12. Lane no 172: “OsGLR3.4”; this should be followed for all GLRs (specifying plant names).
13. Lane no 174: As asked above, how OsGLRs could be manipulated to manipulated to improve the agronomical performance in rice? Add insights.
14. Lane no 184: End of this para, kindly write one line here on what are discussed below.
15. Again there is a lot of opportunity to discuss the insights on aphid control using GLRs which are missed in this section 4.1.
16. Lane no 263: In this sentence, again you use plant names only at a few places, kindly specify the plant names for each GLR throughout the MS.
17. Lane no 177: As asked at several places above, the details on plant names are missing in the gene names such as SOS1, SOS2 and SOS3. Kindly avoid similar kind of mistake in the MS.
18. Lane no 380-384: The authors have written just a one-line insight on the GLRs which needs to be elaborated for each trait in the MS with clear illustrations.
Author Response
Response to reviews
The major revisions have been carefully done in the revised version of the manuscript with tracked changes.
Reviewer 1
General comments:
In the article entitled "Roles of Glutamate Receptor-like Channels (GLRs) in Plant 2 Growth and Response to Environmental Stimuli", the authors reviewed the roles of GLRs in plants. They have listed the various roles of GLRs in plants with reference citations. But this manuscript reads as a compilation or catalog of facts without their critical analysis, thus providing no new ideas about the exploitation of GLRs to improve the agronomic traits of plants.
There is a lot of opportunity to draw inroads with genetic tools like CRISPR/Cas to manipulate the GLR expression to improve the plant performance under adverse conditions, which authors didn't utilize in this manuscript.
Answer:Thank you. Critical analysis has been provided in the corresponding sections of the manuscripts.
Comments:
1. An image illustrating the GLR-mediated Ca2+signaling might benefit the readers.
Answer:An image illustrating the GLR-mediated Ca2+ signaling has been added (Figure 1).
2. Lane no 91: In the sentence "...involved "in" the seed.... "in" is missing!
Answer:Thank you. The writing error has been corrected.
3. Lane no 111:Why are both "was" and "is"?? used in the sentence? The authors should correct it.
Answer:Thank you. The writing error has been corrected.
4. Lane no 113:Here and throughout the manuscript the authors should prefix the GLR names with specific plant names. For eg. AtGLA3.5, would help pinpoint the plant name without the need to write the entire plant name for each sentence. A space saving strategy!
Answer:Thank you. We have prefixed the GLR names with specific plant names throughout the manuscript.
5. Lane no 115:As asked above AtGLR3.1-RNAi?? Kindly specify the plant name for each GLR throughout the MS.
Answer:Have done.
6. Lane no 87-120:Again this para merely compiles the past works without any critical analysis. Kindly add some critical insights on how germination could be increased by manipulating the expression levels of AtGLR3.5, etc.
Answer:Thank you. Critical analysis has been provided in several sections of the manuscripts.
7. Lane no 128:Is it AtGLR3.6??
Answer:Yes. We have prefixed the GLR names with specific plant names throughout the manuscript.
8. Lane no136:"extracellular cells" doesn't seem to be a correct expression!
Answer:Thank you. The error has been corrected.
9. Lane no 146:Why so many "is"? in the sentence. Check and correct it.
Answer:Thank you. The error has been corrected.
10. Lane no 155-162:This para talks a lot about an electrical activity without a link to GLRs! Needs more clear rationale of GLRs here. Again the critical outlook is missing in root development section too.
Answer:Thank you. The paragraph has been deleted, and critical outlook in root development section has been added.
11. Lane no 171:kindly change "osglr3.4" into "Osglr3.4". As asked above, like Osglr, prefix with the plant names for all GLRs.
Answer:Thank you. It has been changed throughout the manuscript.
12. Lane no 172:“OsGLR3.4”; this should be followed for all GLRs (specifying plant names).
Answer:Have done.
13. Lane no 174:As asked above, how OsGLRs could be manipulated to manipulated to improve the agronomical performance in rice? Add insights.
Answer:Thank you. Critical insights has been provided in this section.
14. Lane no 184:End of this para, kindly write one line here on what are discussed below.
Answer:Have done.
15. Again there is a lot of opportunity to discuss the insights on aphid control using GLRs which are missed in this section 4.1.
Answer:Our insights have been added in the section.
16. Lane no 263: In this sentence,again you use plant names only at a few places, kindly specify the plant names for each GLR throughout the MS.
Answer: We have prefixed the GLR names with specific plant names throughout the manuscript.
17. Lane no 177:As asked at several places above, the details on plant names are missing in the gene names such as SOS1, SOS2 and SOS3. Kindly avoid similar kind of mistake in the MS.
Answer:We have prefixed the all the gene or protein names with specific plant names throughout the manuscript.
18. Lane no 380-384: The authors have written just a one-line insight on the GLRs which needs to be elaborated for each trait in the MS with clear illustrations.
Answer:This paragraph has been rewritten.

Reviewer 2 Report
Overall the manuscript is nicely written but required some changes before final acceptance.
English needs improvement throughout the manuscript and the abstract required major corrections as the start of the sentence with 'As' is not a better option.
Figures should be redrawn to give more clarity.
One more section needs to be added that how GLRs interact with plant hormones specially brassinosteroids and salicylic acid.
In my opinion, these points will strengthen the manuscript and will be helpful for further research.
Author Response
Response to reviews
The major revisions have been carefully done in the revised version of the manuscript with tracked changes
Reviewer 2
Overall the manuscript is nicely written but required some changes before final acceptance.
English needs improvement throughout the manuscript and the abstract required major corrections as the start of the sentence with 'As' is not a better option.
Answer:Thank you. We have made thorough revision carefully throughout the manuscript (with tracked changes).
Figures should be redrawn to give more clarity.
Answer:All the figures in the manuscript have been redrawn (Figures 1-3).
One more section needs to be added that how GLRs interact with plant hormones specially brassinosteroids and salicylic acid.
Answer:This section has been added (Section 5).
Round 2
Reviewer 1 Report
The authors have corrected many silly mistakes indicated by this reviewer, for example, prefixing with plant names, minor English corrections, etc. But the major issue of lack of insights and critical outlook for properly utilizing GLRs for crop improvement still exists. The reviewers also suggested adding more insights on AtGLRs, OsGLRs for agricultural trait improvement and especially aphid control have not yet been addressed. As a result, the review still looks like a compilation of past works.
Author Response
Resonse to reviews
Dear editors and reviewers,
Thank you very much for your work on our manuscript. we have provided our insights about GLR utilization in crop improvement in each section of plant growth and environmental stress response including the pest control, and more thinking about the future perspectivewas also added. Additionally, all references have been checked, one by one, for their relevance to the contents of the manuscript. All the revisions are highlighted in revised the manuscript with tracked changes.
If anything else is needed, please don’t hesitate to let me know.
Best regards,
Junli Huang
Round 3
Reviewer 1 Report
The authors have improved the manuscript by adding appropriate insights into the GLRs, which were lacking in the previous versions.